# Differential Expression of Circular RNAs in Rat Brain Regions with Various Degrees of Damage After Ischemia–Reperfusion

**DOI:** 10.3390/ijms262110555

**Published:** 2025-10-30

**Authors:** Ivan V. Mozgovoy, Ekaterina V. Tsareva, Alina E. Denisova, Vasily V. Stavchansky, Leonid V. Gubsky, Lyudmila V. Dergunova, Svetlana A. Limborska, Ivan B. Filippenkov

**Affiliations:** 1Laboratory of Human Molecular Genetics, National Research Center “Kurchatov Institute”, Kurchatov Sq. 2, Moscow 123182, Russia; ivmstalker@gmail.com (I.V.M.); ekaterina.gurianov@yandex.ru (E.V.T.); bacbac@yandex.ru (V.V.S.); dergunova-lv.img@yandex.ru (L.V.D.); limbor.img@yandex.ru (S.A.L.); 2Department of Neurology, Neurosurgery and Medical Genetics, Pirogov Russian National Research Medical University, Ostrovitianov Str. 1, Moscow 117997, Russia; dalina543@gmail.com (A.E.D.); gubskii@mail.ru (L.V.G.); 3Federal Center for the Brain and Neurotechnologies, Federal Biomedical Agency, Ostrovitianov Str. 1, Building 10, Moscow 117997, Russia

**Keywords:** ischemic stroke, RNA-Seq, gene expression, circRNAs, microRNAs, regulatory axes, circRNA-microRNA-mRNA network

## Abstract

Circular RNAs (circRNAs) are non-coding RNAs that can significantly influence the regulation of gene expression in health and disease, including ischemic stroke. We identified 597 differentially expressed circRNAs (DECs) (fold change > 1.5; *Padj* < 0.05) in the striatum region encompassing the ischemic lesion and penumbra 24 h after ischemia–reperfusion injury (tMCAO) in rats, according to high-throughput RNA sequencing data (RNA-Seq). The DECs predominantly increased expression levels relative to those in sham-operated animals. In this study, we also compared these data with DECs we previously identified in the frontal cortex region containing the penumbra and healthy tissue. Furthermore, we bioinformatically constructed a network of competitive circRNA-microRNA-mRNA interactions characterizing the possible functions of DECs in brain areas with varying degrees of ischemic injury. We found that in both tissues, the identified DECs were involved in regulating the expression of genes associated with inflammation and neurotransmission. Moreover, in the striatum, most DECs decreased their expression, while in the frontal cortex, most DECs increased their expression. Thus, we demonstrated different circRNA activities in brain areas with varying degrees of injury. This result may indicate a role for these molecules in regulating brain cell responses, including those important for functional recovery after cerebral ischemia.

## 1. Introduction

Circular RNAs are covalently closed non-coding RNA molecules that play an important role in post-transcriptional regulation of gene expression [1]. It is known that circRNAs compete with mRNA for microRNA binding, thus preventing microRNA-mediated gene expression silencing [2,3]. The involvement of circRNA in the molecular pathogenesis of a number of diseases has been shown, including oncological [4], cardiovascular [5,6], and infectious diseases [7]. In addition, the stability of circRNAs, achieved due to their ring structure, also allows them to be considered as promising diagnostic and prognostic biomarkers [8,9,10,11].

Ischemic stroke remains one of the most socially significant diseases and the second most common cause of death in the world [12]. Ischemic injury has been shown to result in significant metabolomic [13] and transcriptomic [14,15] changes in brain tissue. In recent years, studies have emerged in which authors have studied the expression profile of circRNAs under conditions simulating cerebral ischemia [16,17,18]. Laboratory animal models are widely used in relevant experiments. The transient middle cerebral artery occlusion (tMCAO) model [19] is a well-established model that reflects human stroke events, including the stages of treatment. Specifically, the model reproduces the process of blood flow restoration, both through natural thrombolysis and recanalization and through thrombolytic therapy [19,20].

After a stroke, several pathological zones appear in the brain [21]. The ischemic lesion, or ischemic core, is the region with the most severe damage and is represented mainly by necrotic tissue, while penumbra cells primarily retain the ability to regenerate [22,23]. After ischemic injury, the structure and function of brain tissue change. For example, the formation of glial scars occurs [24]. Changes in the microenvironment and activation of the microglia cause transcriptional, morphological, and metabolic changes in the cells [25].

Previously, using RNA-Seq, we identified more than three thousand differentially expressed genes (DEGs) (fold change > 1.5; *Padj* < 0.05) in the frontal cortex of rats, including the penumbra and undamaged tissue [14], and over four thousand DEGs in the striatum, including the ischemic lesion and penumbra [26], at 24 h after tMCAO. Tissue selection was based on magnetic resonance imaging (MRI) and histological examination. We also identified 64 differentially expressed circRNAs (DECs) in the frontal cortex [27]. The aim of this study was to identify circRNAs involved in the regulation of gene expression in the rat striatum and to compare the DEC profiles in the striatum and frontal cortex, as brain regions with varying degrees of ischemic injury, 24 h after tMCAO. We identified 597 DECs in the striatum. The sets of DECs turned out to differ significantly between the frontal cortex and the striatum, with only 7 DECs common to both brain regions. Furthermore, most of the DECs we detected in the striatum decreased in expression, while in the frontal cortex, most of the DECs increased in expression 24 h after tMCAO. We also constructed a DEC-microRNA-DEG competitive network for the striatum and compared it with a previously proposed network for the frontal cortex [27]. We found that in both tissues, the identified DECs were involved in regulating the expression of genes associated with inflammation and neurotransmission. In the striatum, most DECs showed decreased expression, while in the frontal cortex, they showed increased expression.

Thus, we demonstrated differences in the expression profiles of circRNAs in rat brain regions with varying degrees of ischemic injury 24 h after tMCAO. These differences may indicate the contribution of circRNAs to the formation of a transcriptomic response specific to each brain region and may be significant for cellular recovery after stroke, leading to the development of new treatment strategies.

## 2. Results

### 2.1. Analysis of Differential Expression of circRNAs in Rat Striatum 24 h After tMCAO

Of the 13,890 annotated circRNAs, 597 were differentially expressed (DEC) (fold change > 1.5; *Padj* < 0.05) in the ischemia–reperfusion (IR-s) versus sham-operated (SO-s) animals (Figure 1a). Of these, 403 had decreased expression and 194 had increased expression (Appendix A). The volcano plot (Figure 1b) further illustrates the obtained differential expression results. The most intense increase in expression was observed for *circAdamts9-18.16*, *circCd300lb-3.2*, *circAdamts9-26.22*, *circPvr-4.3*, and *circLtbp1-10.8*. The most intense decrease in expression is for *circMyom2-6.2*, *circTrnau1ap-4.3*, *circTrmt2a-8.7*, *circUsh2a-46.44*, and *circEogt-9.8* (Figure 1c).

Real-time RT-PCR was used to validate the RNA-Seq results. We analyzed circRNAs that, according to RNA-Seq data, showed changes in expression in accordance with the selection criteria (*Padj* < 0.05 and fold change > 1.5): *circMvp-7.4*, which increased expression, and *circPde10a-18.17*, which decreased expression, as well as circRNA that showed no changes in expression (*circPsd2-13.8*) during ischemia. The PCR results for changes in the expression of these molecules relative to *Gapdh* mRNA were consistent with the RNA-Seq results (Figure 1d).

### 2.2. Verification of the Circular Structure of the Studied Transcripts

The primers used for RT-PCR were selected to amplify only the circular transcript, e.g., circRNA. The localization of the primers for *circPde10a-18.17* is shown in Figure 2a. The length of the amplified frontal cortex cDNA fragment during gel electrophoresis (Figure 2b) coincided with that expected for circRNA, as predicted by the PTESFinder program. Sanger sequencing of the PCR product (Figure 2c) further confirmed the presence of a backsplicing site, for *circPde10a-18.17*—between exons 18 and 17. This transcript fragment was deposited in GenBank under accession number PV173812. Thus, the circular structure of *circPde10a-18.17* was confirmed using RT-PCR and subsequent Sanger sequencing (Figure 2d). Sanger sequencing results confirming the backsplicing site in several other circRNAs are presented in Appendix A.

### 2.3. Comparative Analysis of Differential Expression of circRNA and mRNA in the Striatum at 24 h After tMCAO

By mapping the obtained DECs on the rat genome sequences, we found that the 597 DECs obtained in the present study originated from 303 genes. Previously, using RNA-Seq, we identified 4409 genes encoding differentially expressed mRNAs (DEGs) in the striatum 24 h after tMCAO, in the IR-s vs. SO-s comparison [26]. We found that the DEGs included 240 genes for which we detected the DECs they encoded in the present experiment (Figure 3a). For 85 genes (*Adamts9*, *Cdkn1a*, *Dusp10,* and others) that showed increased mRNA expression during ischemia, all of the DECs they encoded also showed increased expression in the IR-s vs. SO-s comparison. At the same time, 155 genes (*Abcb9*, *Rgs9*, *Pde10a*, *Rasgrp2*, etc.) showed a co-directional decrease in mRNA and DEC in IR-s vs. SO-s (Figure 3b, Appendix A). CircRNA *circAdamts9-26.22*, *circAdamts9-27.22*, *circCdkn1a-2.2*, *circAdamts9-30.26*, and *circDusp10-2.2* showed the most intense increase in expression relative to the control (Figure 3b). *CircAbcb9-11.8*, *circRgs9-17.9*, *circPde10a-18.14*, *circPde10a-18.17*, and *circRasgrp2-9.9* showed the most intense decrease in expression (Figure 3b). Sixty-three DECs were also identified whose genes did not change their mRNA expression levels, i.e., they were not DEGs (Figure 3c). For two genes, opposite changes in expression were observed for linear and circular transcripts. Thus, *circCpne4-10.7* circRNA showed an increase in expression, while the mRNA of the *Cpne4* gene encoding it decreased. Similarly, for the *Pank1* gene, an increase in *circPank1-2.2* circRNA expression was observed, but a decrease in mRNA expression was observed (Figure 3d).

### 2.4. Comparative Analysis of Differential Expression of circRNAs in the Striatum and Frontal Cortex of Rats at 24 h After tMCAO

Previously, in an experiment with a similar design, we obtained data on the differential expression of circRNAs in the frontal cortex of the rat brain at 24 h after tMCAO (IR-f vs. SO-f) [27]. We compared this spectrum of DECs, which included 64 circRNAs, with the current results. It was found that 57 DECs were detected exclusively in the frontal cortex, and 590 DECs only in the striatum (Figure 4a, Appendix A). Moreover, 7 circRNAs changed expression in both brain regions (Figure 4b, Appendix A). In all cases, common circRNAs demonstrated codirectional changes in expression. Common DECs were encoded by six genes; the *Elmo1* gene encoded two transcripts that increased expression in both the frontal cortex and the striatum. Among the DECs with increased expression, there were 40 specific for the frontal cortex (*circMast3-8.6*, *circZmiz1-6.4*, *circFam126b-11.7,* and others), 6 common (*circCdkn1a-2.2*, *circAdamts9-27.22*, *circElmo1-13.9*, *circElmo1-13.8*, *circFgfr1-12.10*, *circPpm1h-4.2*), and 188 specific for the striatum (Figure 4c, Appendix A). Among the DECs with decreased expression, there were 17 specific DECs for the frontal cortex, 1 common DEC (*circMast3-8.6*), and 402 specific DECs for the striatum (Figure 4d, Appendix A). Hierarchical cluster analysis of all DECs in IR-s vs. SO-s and IR-f vs. SO-f, as well as their corresponding genes, is presented in Appendix A.

### 2.5. Analysis of the DEC-microRNA-DEG Interaction Network in the Rat Brain Striatum 24 h After tMCAO

Based on the concept of circRNA functioning as competitive endogenous RNAs, we aimed to construct a circRNA-microRNA-mRNA regulatory network. MicroRNA-DEC pairs were predicted (Figure 5a). The possible binding sites of 597 DECs with all 764 microRNAs known for *Rattus Norvegicus* were analyzed. The Miranda program detected 2075 interacting microRNA-DEC pairs, RNAHybrid—7865, TargetScan—69,764. As a result, the intersection of the programs identified 346 microRNA-DEC pairs (Figure 5a). The found pairs included 205 DECs and 150 microRNAs. These microRNAs were selected for further analysis for interactions with DEGs (mRNAs). Thus, microRNA-mRNA interactions were predicted between 4409 DEGs and 150 microRNAs. Miranda detected 40,568 interacting microRNA-DEG pairs, RNAHybrid detected 57,776, and TargetScan detected 298,983. Combination of the results of the three programs contained 8781 microRNA-DEG pairs (Figure 4c), which included 2945 DEGs. Combining microRNA-DEG and microRNA-DEC pairs yielded a competitive DEC-microRNA-DEG network. This network consisted of a total of 3087 axes involving 34 DECs, 45 microRNAs, and 1361 DEGs (Appendix A).

### 2.6. Functional Annotation of DEGs Within the DEC-microRNA-DEG Network in the Rat Brain Striatum 24 h After tMCAO

The list of DEGs within the resulting network was processed using the functional enrichment method using David v 2021. For these DEGs, 140 signaling pathways (KEGG) were obtained with a significance level of *Padj* < 0.05 (Appendix A). Figure 6 compares the obtained signaling pathways with similar results previously obtained for the frontal cortex (IR-f vs. SO-f comparison). Fifty-eight signaling pathways were identified in both comparisons (Figure 6a, Appendix A). The most significant pathways from the intersection included signaling pathways related to neurotransmission—Glutamatergic synapse, Calcium signaling pathway, and others (Figure 6b, Appendix A). Only two signaling pathways stood out in the IR-f vs. SO-f comparison (HIF-1 signaling pathway, Adherens junction). Most of the genes associated with them showed increased expression (Figure 6c, Appendix A).

Among the 82 striatum-specific signaling pathways, pathways associated with inflammation (Ras signaling pathways, Human T-cell leukemia virus 1 infection, etc.) and neurotransmission (GABAergic synapse, Synaptic vesicle cycle, etc.) stood out. DEGs associated with neurotransmission-related pathways mostly showed decreased expression, while DEGs associated with inflammation-related pathways mostly showed increased expression (Figure 6d, Appendix A).

### 2.7. Analysis of the Involvement of circRNAs in the Regulation of Gene Expression Related to the Most Significant Signaling Pathways, Both Common and Specific for the Striatum and Frontal Cortex of Rats 24 h After tMCAO

The obtained signaling pathways were divided into three pathway clusters (PC) based on their representation in the comparisons (Figure 6). Figure 7 shows a network characterizing the involvement of DECs in the presentation of the first three most significant signaling pathways from the PC. Thus, from 82 pathways of PC1, 3 pathways were selected, Osteoclast differentiation, GABAergic synapse, and the Neurotrophin signaling pathway, found only for the DEGs from the network in the IR-s vs. SO-s comparison. PC2 included 2 signaling pathways (Adherens junction, HIF-1 signaling pathway), found only for the DEGs from the network in the IR-f vs. SO-f comparison. PC3 included the three most significant signaling pathways out of 58 (Glutamatergic synapse, Calcium, and MAPK signaling pathways) identified for DEGs from networks constructed for both tissues. For each of these pathways, circRNAs involved in regulating the expression of the maximum number of pathway-associated genes were selected. We found only two circRNAs (*circFgfr1-12.10*, *circAdamts9-27.22*) out of seven common between the IR-f vs. SO-f and IR-s vs. SO-s comparisons, which participated in the formation of the DEC-microRNA-DEG network (Appendix A). Figure 7 is limited to the first 10 circRNAs specific for each of the IR-f vs. SO-f and IR-s vs. SO-s comparisons. It is notable that circRNAs identified as DECs only in IR-f vs. SO-f predominantly decreased expression (*circMegf8-8.6*, *circElmsan1-3.2*, *circWnk2-18.9*, and others), whereas circRNAs identified as DECs only in IR-s vs. SO-s predominantly increased expression (*circSpsb1-2.2*, *circFgfr1-12.9*, *circWdr91-9.2*, and others). CircRNA *circShank3-21.21* competed with the maximum number of DEGs (86) associated with the pathways presented in Figure 7. It is detected only in the IR-s vs. SO-s comparison and decreased expression. This circRNA competed with DEGs associated with pathways from all clusters, PC1–PC3. CircRNA *circWdr91-9.2*, which was upregulated in IR-f vs. SO-f, also competed with the highest number of DEGs (25) from the pathways shown in Figure 7.

We also identified circRNAs that competed with DEGs associated with signaling pathways from only one PC (Appendix A). Specifically, four circRNAs (*circKdm6b-8.3*, *circKcnh3-12.8*, *circTrim2-6.2*, and *circTrim2-6.4*) competed with DEGs involved in the presentation of pathways from PC3 (pathways found for both tissues), and one circRNA (*circCttnbp2-15.9*) competed with DEGs involved in the presentation of pathways from PC1 (pathways found for striatum only). The *circCttnbp2-15.9* DEC was detected in the striatum and is involved in the regulation of striatum-specific pathways (Lipid and atherosclerosis, Inflammatory mediator regulation of TRP channels, Serotonergic synapse) via the *circCttnbp2-15.9*/rno-miR-25-3p/*Cyp2j3* axis (Appendix A). Thus, we characterized the involvement of circRNA in the regulation of gene expression related to the most significant signaling pathways, both common and specific to the striatum and frontal cortex of rats 24 h after tMCAO.

## 3. Discussion

In this study, RNA-Seq was used to examine changes in the circRNA transcriptome in the rat striatum, primarily in the ischemic lesion and penumbra, 24 h after tMCAO. Among the 13,890 circRNAs identified, 597 are differentially expressed (DECs) in the rat striatum 24 h after tMCAO. Most of them (403 DECs) show decreased expression. We previously demonstrated a decrease in circRNA level in subcortical brain structures after tMCAO with long-term anesthesia [28]. On the one hand, downregulation of circRNAs may be associated with a decrease in the number of surviving cells in the ischemic area. On the other hand, this may be one manifestation of transcriptomic changes in response to injury in the striatum and due to the use of circRNAs in regulatory networks. Among the downregulated circRNAs is *circShank3-21.21*, which has been shown to stimulate neuroinflammation by competing with *TLR4* mRNA [29]. Some of the DECs we identified are encoded by genes such as *Phactr1* [30], *Fgfr1* [31], and others that have been shown to be associated with stroke.

We also compared our data for the rat striatum with the set of DECs we had previously identified in the frontal cortex region containing the penumbra and healthy tissue [27]. This comparison reveals both common and specific DECs associated with brain areas with varying degrees of ischemic injury. The comparison revealed that the number of DECs identified in the striatum (597) significantly exceeded the number of DECs identified in the frontal cortex (64) 24 h after tMCAO. Moreover, most DECs in the frontal cortex show increased expression, while most DECs in the striatum show decreased expression. A total of 7 circRNAs (*circCdkn1a-2.2*, *circAdamts9-27.22*, *circElmo1-13.9*, *circElmo1-13.8*, *circFgfr1-12.10*, *circPpm1h-4.2*, *circMast3-8.6*) detected changed expression in both brain areas. It is possible that they may persist as the most stable markers of ischemic damage. It is noteworthy that circRNA of the *Cdkn1a* gene, encoding the p21 protein, is detected in both areas. This protein plays an extremely important role in the regulation of the cell cycle [32], and the mRNA encoding it has been described as a biomarker of oxidative stress and apoptosis in stroke [33]. The *Mast3* gene, encoding *circMast3-8.6*, is involved in the development of the nervous system, and its role in CNS pathologies is actively studied [34,35,36].

Most of the circRNAs we detected as differentially expressed in the striatum of ischemic rats were not detected in the frontal cortex. For example, expression of six circRNAs of the *Vwf* gene, encoding von Willebrand factor, was reduced exclusively in the striatum. Increased expression of this gene is known to increase the risk of thrombosis and, consequently, the risk and severity of stroke [37,38].

Transcriptomic changes landscape of complex pathological conditions such as stroke can be mediated, in part, by network interactions between RNA molecules, both coding and non-coding. The changes in circRNA levels, along with changes in mRNA levels, during ischemia may be related to the role of circRNAs as microRNA “sponges”, protecting mRNA [39]. It is currently known that each microRNA can interact with the transcripts of a large number of genes [40,41,42,43]. We note that similar mechanisms can also be implemented in the striatum, which predominantly contains the ischemic lesion. Using bioinformatics approaches, we predicted probable interactions between microRNAs and mRNAs and microRNAs and circRNAs using three independent algorithms: Miranda, RNAHybrid, and TargetScan. We demonstrated that 205 DECs and 2945 DEGs are involved in this regulatory network in the striatum, and 115 microRNAs mediate competition between them.

Of the seven DECs we identified in both the striatum and the frontal cortex of rats 24 h after tMCAO, two circRNAs, namely *circFgfr1-12.10* and *circAdamts9-27.22*, are found to be involved in competitive DEC-microRNA-DEG axes. These circRNAs form a total of 206 axes common to both brain zones. Among these axes, for example, is the *circFgfr1-12.10*/rno-miR-665/*Fgfr1* axis. Moreover, in both zones, the expression of both *circFgfr1-12.10* circRNA and the mRNA of the fibroblast growth factor receptor 1 gene (Fgfr1), which encodes this circRNA, is increased. It is known that the level of *Fgfr1* mRNA expression increases under ischemic conditions [44]. The FGFR1 protein is involved in maintaining the blood-brain barrier during ischemia and can stimulate angiogenesis and revascularization of the affected area [31,44]. Thus, protection of *Fgfr1* gene activity with the help of circRNAs can promote cellular restoration in tissues after ischemic injury, including in the penumbra zone.

Some of the microRNAs in the network have previously been described as associated with stroke. For example, increased expression of microRNA-3473b has been shown to have a pro-inflammatory effect, and artificially reducing its expression led to a decrease in lesion size and the concentration of inflammatory factors in the striatum [45]. We found that circRNAs that interact with miR-3473b (*circGli3-11.5* and *circGli3-11.7*) are downregulated in the striatum of ischemic rats. Axes involving miR-3473b were detected only in the striatum. However, we detected a total of 21 microRNAs involved in regulatory axes in both the frontal cortex and striatum. In both areas, we predicted axes involving miR-615-5p, which has been shown to play a role in the repair of vascular damage [46]. This microRNA has been proposed as a potential stroke biomarker [47].

Construction of competitive networks of DEC-microRNA-DEGs and subsequent functional annotation of the DEGs within the networks allowed us to assess the potential contribution of circRNA-mediated transcriptome regulation during ischemia in different brain regions. We demonstrated that in both the striatum and frontal cortex, 24 h after tMCAO, genes potentially regulated by circRNAs are primarily related to neuroinflammation (Proteoglycans in cancer, MAPK signaling pathway) and neurotransmission (Glutamatergic synapse Calcium, cAMP signaling pathway) (Figure 8). We grouped the corresponding signaling pathways into PC3. Most genes associated with inflammatory signaling pathways are upregulated in both regions, while most genes associated with neurosignaling are downregulated. Furthermore, a specific spectrum of pathways is observed for each tissue. In particular, a cluster of 82 pathways (PC1) is found in the striatum, which are also associated with neurotransmission and inflammation systems, but are not found in the frontal cortex. Interestingly, the cluster of pathways specific to the frontal cortex (PC2) includes only 2 pathways (Adherens junction, HIF-1 signaling pathway). This may be due to a less intense transcriptomic response of circRNAs in the frontal cortex (64 DECs) compared to the striatum (597 DECs). The majority of circRNAs identified in tissues are involved in the regulation of the expression of genes associated with pathways from all clusters (PC1–PC3). Among them is the circRNA of the *Shank* gene (*circShank3-21.21*), associated with the most significant signaling pathways from PC1–PC3. We previously noted the high representation of *Shank* family genes in the network we obtained for the frontal cortex, as well as the known role of these genes in the formation and maintenance of synapse structure [48,49]. We also identified circRNAs that compete with DEGs associated with signaling pathways from only one PC. Thus, *circCttnbp2-15.9*, which had changed expression only in the striatum, was involved in the presentation of pathways characteristic only of the striatum (PC1). We predicted the *circCttnbp2-15.9*/rno-miR-25-3p/*Cyp2j3* axis, which is responsible for the implementation of the activity of this circRNA in the striatum. It is noteworthy that the *Cttnbp2* gene is involved in the formation of the dendritic cytoskeleton, and knockdown of this gene disrupts neurogenesis in mice [50]. We found that *circCttnbp2-15.9*, like the mRNA of this gene, decreases expression in the striatum, which may be one of the manifestations of transcriptomic changes in response to injury. We also found decreased expression of the *Cyp2j3* gene. There is evidence that decreased expression of this gene is characteristic of neuroinflammation [51].

A limitation of our study is the use of computational tools alone to identify circRNA–miRNA–mRNA networks, so the functional validation of these networks remains entirely in silico. Also, microRNA expression was not measured for this study, so a network was constructed with all known *R. norvegicus* microRNAs. Additionally, there was a lack of different timepoints to provide the dynamic changes of circRNA expressions, as well as single-cell RNA-Seq experiments. Furthermore, the use of animal models imposes additional limitations on the translation of results due to the animal–human species barrier. Although the rat tMCAO model reflects human stroke events, the use of human cell cultures may be useful in the future. Cellular cultures allow for the modeling of individual stages of the ischemic cascade and facilitate the use of genetic engineering approaches, which are essential for studying the functions of genes and circRNAs in stroke pathogenesis.

Thus, in this study, we characterize the involvement of circRNAs in the regulation of gene expression related to the most important signaling pathways, both general and specific to the striatum and frontal cortex of rats, containing cells with varying degrees of damage 24 h after tMCAO. We demonstrate that different circRNA activities are observed in the studied brain regions. These differences may indicate a special role for circRNAs in regulating cellular responses to pathological injury, including those important for restoring cell function in the penumbra.

## 4. Materials and Methods

### 4.1. Experimental Animals

Male Wistar rats (2 months old, 200–250 g) were obtained from AlCondi, Ltd. (Moscow, Russia). Animals were contained in groups of 4–5 per M-6 cage under a 12-h light/dark cycle at 22–24 °C with free access to food and water. Animals were divided into sham-operated (SO) and ischemia–reperfusion (IR) groups randomly. During subsequent sample preparation, RNA sequencing, and initial data processing, the experimenter was blinded to the experimental or control groups to avoid bias in the analysis. The experimental design is presented in Appendix A.

### 4.2. tMCAO Model

The tMCAO model was performed according to the method of Koizumi et al. [52] with modifications described previously [53]. tMCAO details are described in Appendix A. The model was performed under MRI guidance using a ClinScan small animal CT scanner (Bruker BioSpin, Billerica, MA, USA) with a magnetic field induction of 7 T. Rats were anesthetized with isoflurane during magnetic resonance imaging (MRI). In all rats of the IR-s group, ischemic damage was localized in the ipsilateral hemisphere of the rat brain 24 h after tMCAO (MRI data). The damaged area extended to the subcortical region and part of the cortex, with the lesion localized in the striatum. The rats of the “sham operation” (SO) group were anesthetized and subjected to a similar surgical procedure (neck incision and separation of the bifurcation, anesthesia), but without arterial occlusion.

Each experimental group consisted of 5 animals. Rats were anesthetized and decapitated 24 h after tMCAO (IR-s group) and sham surgery (SO-s group). All rats were alive before decapitation (24 h after tMCAO).

### 4.3. RNA Isolation

The striatum was removed and stored in RNAlater solution (Ambion, Austin, TX, USA) for 24 h at 4 °C and then stored at −70 °C. Total RNA was isolated using TRI Reagent (MRC, Cincinnati, OH, USA) according to the manufacturer’s recommendations. The isolated RNA was treated with DNase I (Thermo Fisher Scientific Baltics UAB, Vilnius, Lithuania) in the presence of RiboLock RNase inhibitor (Thermo Fisher Scientific Baltics UAB, Vilnius, Lithuania), according to the manufacturer’s recommended protocol. RNA integrity was verified with capillary electrophoresis (Experion, BioRad, Hercules, CA, USA). The RNA integrity number (RIN) was at least 9.0.

### 4.4. RNA-Seq

In this experiment, we used total RNA isolated from the striatum, including the ischemic lesion. RNA sequencing was performed with the participation of Genoanalitika LLC, Moscow, Russia. The circRNA fraction (circRNA) was obtained using the PureLink RNA Micro kit (Invitrogen, Thermo Fisher Scientific Baltics UAB, Vilnius, Lithuania) and the Trizol reagent (Thermo Fisher Scientific Baltics UAB, Vilnius, Lithuania). The isolated RNA was treated with DNase I (Thermo Fisher Scientific Baltics UAB, Vilnius, Lithuania) in the presence of the ribonuclease (RNase) inhibitor RiboLock (Thermo Fisher Scientific Baltics UAB, Vilnius, Lithuania). Ribosomal RNA was removed using the RiboMinus Eukaryote KIT (Ambion, Thermo Fisher Scientific, Waltham, MA, USA), and the obtained RNA was treated with RNase R (Lucigen, Middleto, WI, USA), according to the manufacturer’s recommendations. cDNA libraries for circRNAs were prepared using NEB Next^®^Ultra™ II RNA Library Prep (New England Biolabs, Ipswich, MA, USA). The concentration of cDNA libraries was measured using Qbit 2.0 and the Qubit dsDNA HSAssay Kit (Thermo Fisher Scientific). The fragment length distribution of the library was determined using the Agilent High Sensitivity DNAKit (Agilent, Santa Clara, CA, USA). Sequencing was performed using an Illumina HiSeq 2500 (Illumina, San Diego, CA, USA). At least 20 million reads were generated.

### 4.5. RNA-Seq Data Analysis

CircRNA reads were mapped to the rat genome (rno6 version) using Bowtie2 [54]. circRNA sequences were assembled using PTESFinder [55]. Differential expression analysis of circRNAs in IR-s versus SO-s was performed using the DESeq2 package, which allows using variation information from the entire gene set to estimate overall variance and correct for gene-specific variance with a small number of replicates [56]. Each comparison group included three animals (Appendix A). Only circRNAs that showed expression changes greater than 1.5-fold with a Benjamini–Hochberg-corrected *p*-value (*t*-test) below 0.05 (*Padj* < 0.05) were considered as DECs.

### 4.6. cDNA Synthesis

cDNA synthesis was performed in a 20 μL reaction mixture containing 5 μg of total RNA using the RevertAid First Strand cDNA Synthesis Kit reagents (Thermo Fisher Scientific Baltics UAB, Vilnius, Lithuania) according to the manufacturer’s instructions.

### 4.7. Real-Time Reverse Transcription Polymerase Chain Reaction (RT-PCR)

Synthesized cDNA was used as a template for real-time PCR with the intercalating dye SYBR Green I. Primers were designed using the Oligo Analyzer Tool (https://www.idtdna.com/pages/tools/oligoanalyzer (accessed on 1 July 2025)) and synthesized by Evrogen. Primers are presented in Appendix A. cDNA amplification was performed using a StepOnePlus™ Real-Time PCR System (Applied Biosystems, Thermo Fisher Scientific, Waltham, MA, USA).

Each group included five animals (Appendix A). Each cDNA sample was analyzed in triplicate. The average threshold cycle (Ct) value was calculated from three replicate measurements. The mRNA level of the glyceraldehyde-3-phosphate dehydrogenase (*Gapdh*) gene was used to normalize the PCR results. Calculations were performed using the Relative Expression Software Tool 2005 software (REST, gene-quantification, Freising-Weihenstephan, Bavaria, Germany) [57]. The manual at the site ‘REST.-gene-quantification.info’ was used to evaluate the expression of target genes relative to the expression levels of the reference genes. To estimate the content of the studied circRNAs relative to the mRNA of the comparison genes, the formula 2ΔCt was used, where ΔCt = Ct(tar) − Ct(ref), Ct(ref) is the average Ct value for the studied circRNAs, and Ct(ref) is the average Ct value for the mRNA of the comparison gene (*Gapdh*). When comparing data groups, differences with a probability value of *p* < 0.05 (two-sided Pair-Wise Fixed Reallocation Randomization Test) were considered statistically significant. Additional data processing was performed using Microsoft Excel (Microsoft Office 2010, Microsoft, Redmond, WA, USA).

### 4.8. Electrophoresis of PCR Products

PCR products were extracted with a mixture of chloroform and isoamyl alcohol (24:1). DNA was dissolved in 9 μL of buffer (10 mM Tris (pH 9.0), 50 mM sodium chloride), 1 μL of 10× dye (50 mM Tris (pH 8.27), 0.25% bromophenol blue, 60% glycerol) and loaded onto a horizontal agarose gel (2% agarose, 1× TAE, 0.6 μg/mL ethidium bromide). Electrophoresis was performed in 1× TAE buffer for 35 min in a horizontal electrophoresis chamber (BioRad, Hercules, CA, USA) at 5 V/cm (Elf-8 power supply, DNA-technology, Moscow, Russia).

### 4.9. Sanger Sequencing of PCR Products

Sanger sequencing of PCR products. Individual bands were excised from the agarose gel. Subsequent DNA extraction from the gel and Sanger sequencing of the PCR products were performed at Evrogen, Moscow, Russia. Sanger sequencing results were analyzed using Chromas Lite software, Version 2.6.6 (Technelysium Pty Ltd., Brisbane, Australia).

### 4.10. Bioinformatic Identification of circRNA-microRNA-mRNA Networks

Three programs were used to predict interactions between RNAs: Miranda v.3.3 [58], RNAhybrid v.2.2.1 [59], and TargetScan v.8 [60]. Miranda and RNAHybrid were used with the following arguments:

miranda <input_microRNAs.fasta> <input_targets.fasta> -sc155 -en -25 -go -9 -ge-4

RNAhybrid -m 5000 -d -c -e 25 -b 100 -t <input_targets.fasta> -q <input_microRNAs.fasta>

where for Miranda: sc—minimum score; en—maximum ΔG of the duplex; go—gap opening penalty; ge—gap extension penalty; for RNAHybrid: m—maximum length of the target RNA; e—maximum ΔG of the duplex; b—maximum number of microRNAs interacting with the target RNA.

Mature microRNA sequences were retrieved from the MirBase database [61] according to recommendations [62]. The interactions predicted by the three tools were then considered. Cytoscape 3.9.1 (Institute for Systems Biology, Seattle, WA, USA) was used to visualize the resulting network.

### 4.11. Functional Analysis

Functional analysis was performed for genes whose mRNAs are involved in the circRNA-microRNA-mRNA network using David v2021 [63]. Only signaling pathways identified with a significance level of *Padj* < 0.05 (with the Benjamini–Hochberg correction) were considered for further analysis.

### 4.12. Hierarchical Cluster Analysis

Hierarchical cluster analysis was performed using the Heatmapper platform (http://heatmapper.ca/ (accessed on 1 July 2025)) [64]. Other calculations and plots, including Volcano-plot graphs, were generated using Microsoft Excel.

### 4.13. Availability of Data and Material

RNA-Seq data have been deposited in the Sequence Read Archive database under the accession code PRJNA1151256 (SAMN43308746-SAMN43308751) [65], (circRNA from frontal cortex); PRJNA1119923 (SAMN41664920-SAMN41664925) [66], (mRNA from frontal cortex); PRJNA1151230 (SAMN43305801-SAMN43305806) [67], (circRNA from striatum), PRJNA1151256 (SAMN41664920-SAMN41664925) [68], (mRNA from striatum).

## 5. Conclusions

In this study, high-throughput RNA sequencing (RNA-Seq) was used to obtain a spectrum of circRNAs differentially expressed in the rat brain striatum 24 h after tMCAO, while the striatum included the extensive ischemic core and penumbra. Using bioinformatics tools, a network of circRNA-microRNA-mRNA competitive interactions was identified. Comparison of the DEC spectra and the structure of the competitive networks revealed that circRNA activity differs in brain areas with different degrees of ischemic injury. These differences may indicate the involvement of circRNAs in regulating the transcriptomic response of cells to pathological injury, including in the infarct and penumbra zones. Furthermore, we demonstrated that genes whose expression may be dependent on circRNA activity are responsible for synaptic signaling and the inflammatory response, confirming previous data. Our study demonstrates the significant role of circRNA-mediated regulation of the brain transcriptome under ischemic conditions and allows us to consider circRNAs as potential targets for new strategies for stroke prevention and treatment of post-stroke complications.

## Figures and Tables

**Figure 1 ijms-26-10555-f001:**
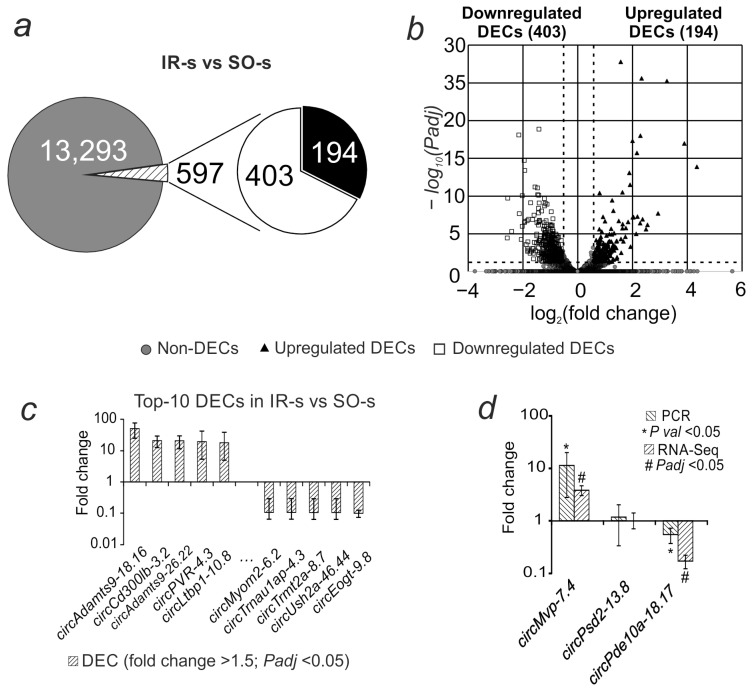
Differential expression of circRNAs in rat striatum 24 h after tMCAO. (**a**) The total number of annotated circRNAs is shown as a gray circle; the numbers of upregulated and downregulated DECs in the IR-s vs. SO-s comparison (fold change > 1.5; *Padj* < 0.05) are shown as white and black sectors of the circle, respectively. (**b**) Volcano plot of RNA-Seq results. (**c**) The first 10 DECs that showed the greatest change in expression in the IR-s vs. SO-s comparison. (**d**) PCR verification of RNA-Seq results. Each PCR data comparison group includes 5 animals; each RNA-Seq data comparison group includes 3 animals. Data are presented as the mean ± SEM. ΔCt = Ct(tar) − Ct(ref), Ct(ref) is the average Ct value for the studied circRNAs and Ct(ref) is the average Ct value for the mRNA of the comparison gene (*Gapdh*).

**Figure 2 ijms-26-10555-f002:**
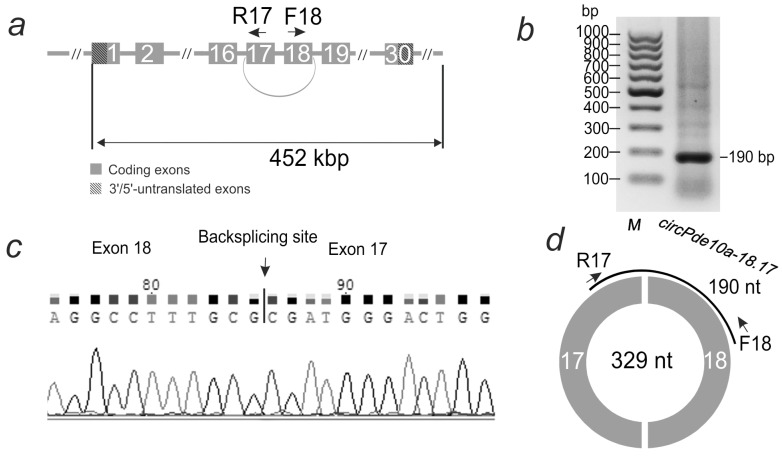
Analysis of circRNA structure in rat striatum 24 h after tMCAO. (**a**) Structure of the *Pde10a* gene. Exons are shown as grey rectangles. Exons 18 and 17, connected by an arc, are involved in the formation of *circPde10a*-*18.17* circRNA. Primers (F-forward, R-reverse) and their direction are shown by arrows. (**b**) Electrophoresis of the product of the PCR reaction carried out with these primers. On the left—the position of the GeneRuler 100 bp DNA Ladder marker (Thermo Fisher Scientific, Baltics UAB, Vilnius, Lithuania). (**c**) A fragment of Sanger sequencing results of the PCR product, including the backsplice site between exons 18 and 17, according to the results of Sanger sequencing. (**d**) Structure of *circPde10a-18.17* circRNA. The *Pde10a* gene exons incorporated into the circRNA are shown as white numbers on the circle segments. The number inside the circle is the length of the circRNA, and PCR amplicon is shown as an arc.

**Figure 3 ijms-26-10555-f003:**
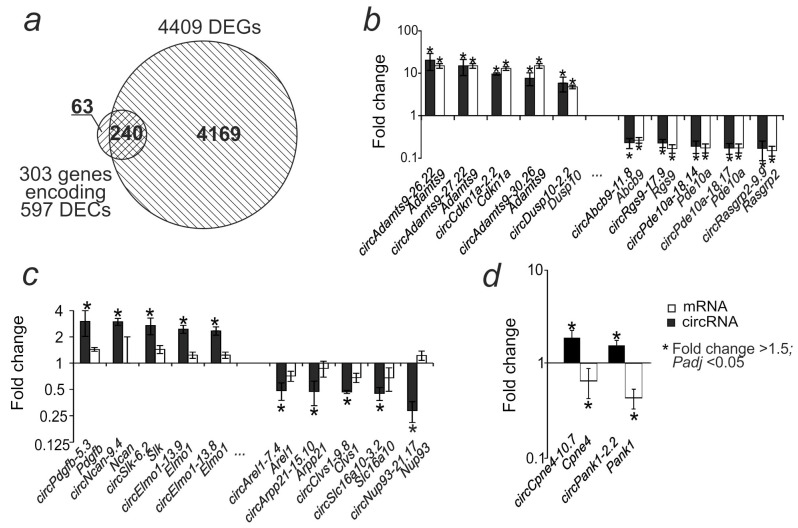
Comparative analysis of differential expression of circRNA and mRNA in the striatum 24 h after tMCAO. (**a**) The intersection of the DEC-encoding genes obtained in this study with the DEGs identified previously is shown using a Venn diagram. (**b**) The first 10 DECs with the highest fold change in expression and the corresponding DEGs from the intersection in the Venn diagram. (**c**) The first 10 DECs with the highest fold change in expression, the genes of which were not significant as DEGs. (**d**) Genes for which mRNA and circRNA showed oppositely directed expression changes.

**Figure 4 ijms-26-10555-f004:**
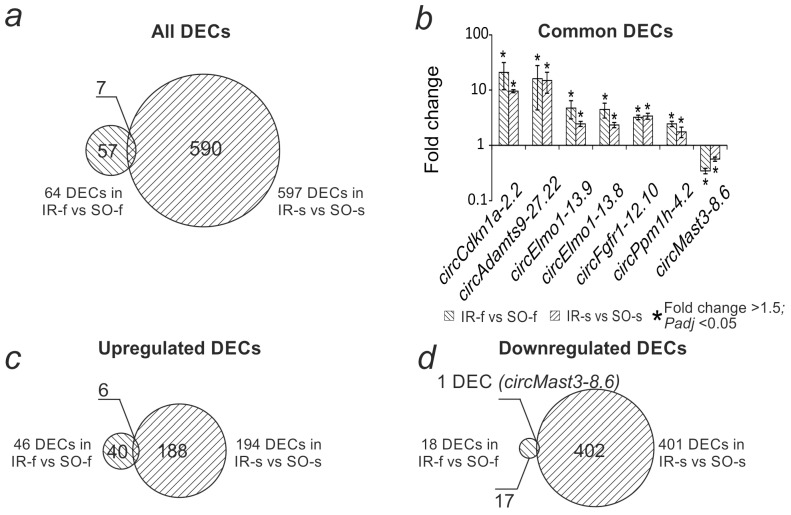
Comparative analysis of differential expression of circRNAs in the striatum (IR-s vs. SO-s) and frontal cortex (IR-f vs. SO-f) 24 h after tMCAO. (**a**) Venn diagram showing the intersection of the spectra of DECs in the striatum (IR-s vs. SO-s) and frontal cortex (IR-f vs. SO-f). (**b**) Changes in DEC expression levels found in both comparisons. (**c**) Venn diagram showing the intersection of the spectra of increased DEC expression in the striatum and frontal cortex. (**d**) Venn diagram showing the intersection of the spectra of decreased DEC expression in the striatum and frontal cortex.

**Figure 5 ijms-26-10555-f005:**
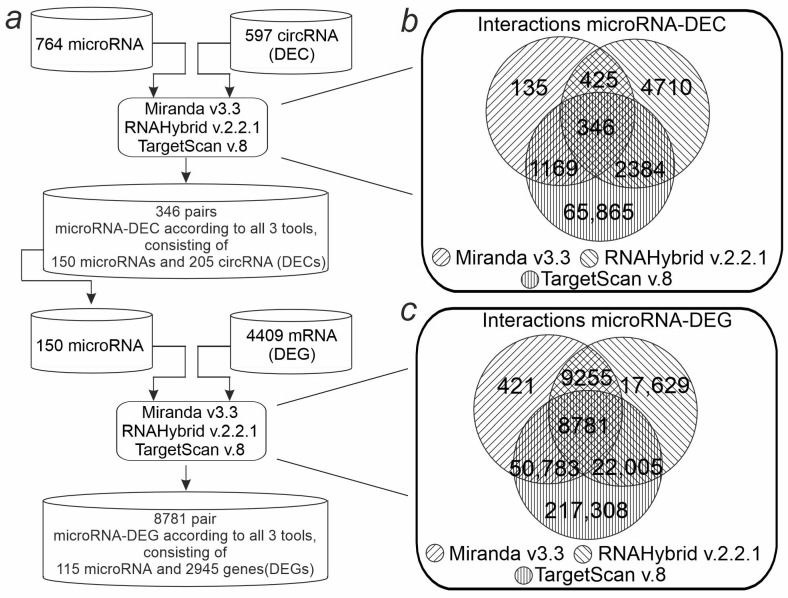
Bioinformatic pipeline for identification of the DEC-microRNA-DEG network in the striatum. (**a**) Schematic representation of the workflow for network identification. (**b**) Venn diagram showing the number of common and specific microRNA-DEC pairs according to the results from three programs (Miranda, RNAHybrid, TargetScan). (**c**) The number of common and unique microRNA-DEG pairs according to the results from three programs is shown using the Venn diagram.

**Figure 6 ijms-26-10555-f006:**
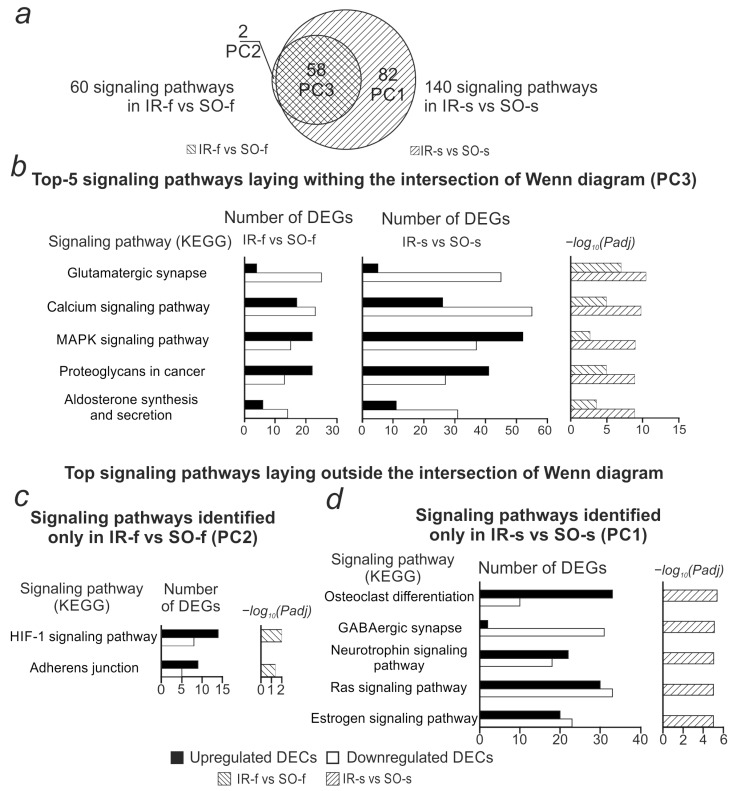
Comparison of signaling pathways associated with genes involved in the circRNA-microRNA-mRNA network in the IR-f vs. SO-f and IR-s vs. SO-s comparisons. (**a**) Venn diagram showing the number of common and specific signaling pathways identified in the IR-f vs. SO-f and IR-s vs. SO-s comparisons. (**b**) Top 5 signaling pathways by significance level (*Padj*) identified for both IR-f vs. SO-f and IR-s vs. SO-s comparisons. (**c**) Signaling pathways identified for the IR-f vs. SO-f comparison, but not IR-s vs. SO-s. (**d**) Top 5 pathways by significance level identified for the IR-s vs. SO-s comparison, but not IR-f vs. SO-f. For each pathway, the number of up- and down-regulated DEGs associated with that pathway and the pathway significance level (*Padj*, Benjamini–Hochberg correction) are shown.

**Figure 7 ijms-26-10555-f007:**
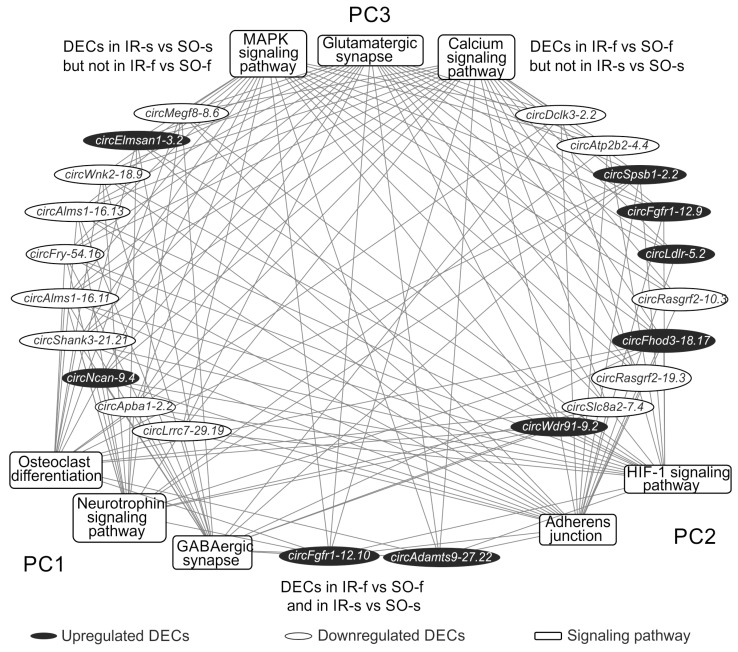
A fragment of the DEC-signaling pathway network obtained for the striatum and frontal cortex at 24 h after tMCAO. Signaling pathways (KEGG) are shown as rectangles, DECs as ovals. Signaling pathways are grouped into clusters. Pathway cluster 1 (PC1) includes signaling pathways that are distinguished only in the IR-s vs. SO-s comparison. Pathway cluster 2 (PC2) includes signaling pathways that are distinguished only in the IR-f vs. SO-f comparison. Pathway cluster 3 (PC3) includes signaling pathways that are distinguished for both comparisons, IR-f vs. SO-f and IR-s vs. SO-s. The first 3 most significant signaling pathways from PC1–PC3 are shown. DECs on the right side of the figure are found only in the IR-s vs. SO-s comparison; on the left side of the figure, only in the IR-f vs. SO-f comparison. The first 10 DECs involved in competition with the maximum number of genes (mRNAs) associated with PC2–PC3 are shown. Only two DECs involved in pathway presentation are found in both IR-f vs. SO-f and IR-s vs. SO-s comparisons, are shown at the bottom of the figure. Lines connecting DECs and signaling pathways indicate that DECs compete with DEGs associated with pathways.

**Figure 8 ijms-26-10555-f008:**
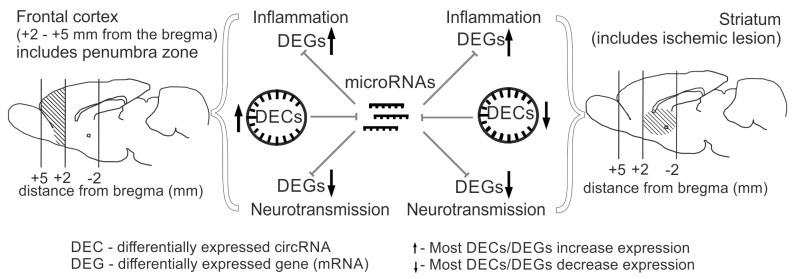
Graphical representation of RNA-Seq results obtained for the frontal cortex and striatum of rat brains 24 h after tMCAO. The results show that both brain tissues exhibit increased expression of inflammatory genes and decreased expression of neurotransmission genes, while circRNA levels are predominantly increased in the frontal cortex and decreased in the striatum. CircRNAs may potentially be involved in controlling the expression of genes involved in inflammation and neurotransmission by regulating microRNA activity in both brain tissues. Brain regions analyzed are hatched.

## Data Availability

The datasets used in this study can be found here: circRNAs from frontal cortex [65], mRNAs from frontal cortex [66], circRNAs from striatum [67], mRNAs from striatum [68].

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
