# Peer review of "Differential Expression of Circular RNAs in Rat Brain Regions with Various Degrees of Damage After Ischemia–Reperfusion"

_ijms, 2025, doi:10.3390/ijms262110555_

Round 1

Reviewer 1 Report

Comments and Suggestions for Authors

The manuscript “Differential Expression of Circular RNAs in Rat Brain Regions with Various Degrees of Damage after Ischemia-Reperfusion” presents a well-designed study with a solid experimental basis, appropriate data interpretation, and only minor writing issues.

We considered that it is suitable for publication after a minor revision focused on improving textual clarity and consistency in academic English.

The study is scientifically sound, correctly interpreted, and well contextualized within the current literature. The conclusion—that circRNA-mediated regulatory patterns differ according to the degree of ischemic brain damage—is coherent with the data presented.

This is a technically robust and timely contribution to the field of transcriptomics in ischemic brain injury.

The authors employ a validated rat model of transient middle cerebral artery occlusion (tMCAO), effectively differentiating between the striatum (injury core) and the frontal cortex (penumbra/healthy region).

The methodology is rigorous, combining RNA-Seq with RT-qPCR validation and multi-platform bioinformatics (miRanda, RNAhybrid, TargetScan).

A total of 597 differentially expressed circRNAs (DECs) were identified in the striatum and 64 in the cortex, forming the basis for a functional circRNA–miRNA–mRNA network. The authors successfully link these findings to inflammation and neurotransmission pathways, highlighting several circRNAs as potential modulators of post-ischemic molecular responses.

Minor

As the authors indicate in the manuscript:

Only a single post-injury time point (24 h) was analyzed; therefore, no conclusions can be made about the temporal evolution of molecular changes, only about their state at that specific moment (24 hours post-injury).

The sample size (n=3 for RNA-Seq, n=5 for RT-qPCR) is modest though acceptable; a brief discussion of statistical power would strengthen the manuscript.

Functional validation remains entirely in silico.

Additional Suggestions

Abstract: Lines 19–27 present an apparent inconsistency (“DECs predominantly increased expression…” vs. “most DECs decreased…”). Clarify whether upregulation or downregulation predominates in each region.

Abbreviations: Reintroduce abbreviations (DEC, DEG, tMCAO) when starting new sections.

Verb tenses: Use the past tense in Results (“we found,” “we observed”) and the present tense in Discussion (“our results suggest”).

Formatting:

Remove extra spaces before numerical citations [XX].

Avoid periods after numerals in author names.

Ensure consistent typography and spacing.

Comments on the Quality of English Language

The English could be improved to more clearly express the research.

Author Response

Response to the comments of Reviewer 1 to Manuscript ID: ijms-3947956

Authors:

We are very grateful to the Reviewer 1 for the review and constructive comments. We carefully considered the comments of the Reviewer 1 and attached the answers to all comments.

Reviewer 1:

  1. Only a single post-injury time point (24 h) was analyzed; therefore, no conclusions can be made about the temporal evolution of molecular changes, only about their state at that specific moment (24 hours post-injury).

Authors:

In accordance with the Reviewer’s recommendation, changes were added in the text of the Manuscript as a limitation of our study (lines 418-420 in Mark-up Copy_R1).

Reviewer 1:

  1. The sample size (n=3 for RNA-Seq, n=5 for RT-qPCR) is modest though acceptable; a brief discussion of statistical power would strengthen the manuscript.”.

Authors:

In Methods section we provide a rationale for the applicability of our methods (namely, the differential expression analysis package) DESeq2 to this sample size, referring to the paper by DESeq2 authors, where the sample size is also discussed (lines 487-491 in Mark-up Copy_R1).

Reviewer 1:

  1. Functional validation remains entirely in silico.

Authors:

Changes were added in the text of the Manuscript as a limitation of our study (lines 416-418 in Mark-up Copy_R1).

Reviewer 1:

  1. Abstract: Lines 19–27 present an apparent inconsistency (“DECs predominantly increased expression…” vs. “most DECs decreased…”). Clarify whether upregulation or downregulation predominates in each region.

Authors:

In accordance with the Reviewer’s recommendation, changes were added in the text of the Manuscript (line 28 in Mark-up Copy_R1).

Reviewer 1:

  1. Abbreviations: Reintroduce abbreviations (DEC, DEG, tMCAO) when starting new sections.

Authors:

In accordance with the Reviewer’s recommendation, changes were added throughout the text (file “Mark-up Copy_R1”).

Reviewer 1:

  1. Use the past tense in Results (“we found,” “we observed”) and the present tense in Discussion (“our results suggest”).

Authors:

In accordance with the Reviewer’s recommendation, changes were added throughout the text (file “Mark-up Copy_R1”).

Reviewer 1:

  1. Remove extra spaces before numerical citations [XX].

Authors:

In accordance with the Reviewer’s recommendation, changes were added throughout the text (file “Mark-up Copy_R1”).

  1. Avoid periods after numerals in author names.

Authors:

In accordance with the Reviewer’s recommendation, changes were added throughout the text (lines 4-5 in Mark-up Copy_R1

  1. Ensure consistent typography and spacing.

Authors:

In accordance with the Reviewer’s recommendation, changes were added throughout the text (file “Mark-up Copy_R1”).

Reviewer 2 Report

Comments and Suggestions for Authors
  1. In Figure 2, QC for only circRNA gene is shown. Can you include more QC examples in the supplementary for other circRNAs? Eg. an excel sheet with the the Sanger sequencing results which include the backspacing sites.
  2. In the section with lines 187-188, please include the rationale for finding the circRNA-microRNA and microRNA-DEGs pairs. This will make the biological relevance of this network analysis more clear. Some rationale is provided in the discussion but please include more information here as well.
  3. Why were all the 764 microRNAs known for Rattus Norvegicus analyzed? These may not be expressed in the ischemic stroke model in the brain regions analyzed and the obtained network analysis results may not be present in the brain regions. Please include the rationale for including all microRNAs or include some more information in the discussion regarding this. 
  4. Why does targetscan yield significantly higher number of interactions compared to the other techniques used? Please include any limitations with this method.
  5. Line 292- Most of them (403 DECs) showed decreased expression. Please discuss a hypothesis, referring to previous studies, why this might be the case after an ischemic stroke and how this is important to the transcriptomic changes observed after stroke. This provides more biological relevance to the results observed in this study. 

Author Response

Response to the comments of Reviewer 2 to Manuscript ID: ijms-3947956

Authors:

We are very grateful to the Reviewer 2 for the review and constructive comments. We carefully considered the comments of the Reviewer 2 and attached the answers to all comments.

Reviewer 2:

  1. In Figure 2, QC for only circRNA gene is shown. Can you include more QC examples in the supplementary for other circRNAs? Eg. an excel sheet with the the Sanger sequencing results which include the backspacing sites.

Authors:

In accordance with the Reviewer’s recommendation, Sanger sequencing results of 7 circRNAs were added as Supplementary Table S2 and mentioned in manuscript (lines 124-125 in Mark-up Copy_R1).

Reviewer 2:

  1. In the section with lines 187-188, please include the rationale for finding the circRNA-microRNA and microRNA-DEGs pairs. This will make the biological relevance of this network analysis more clear. Some rationale is provided in the discussion but please include more information here as well.

Authors:

In accordance with the Reviewer’s recommendation, changes were added in the text of the Manuscript (lines 197-199 in Mark-up Copy_R1).

Reviewer 2:

  1. Why were all the 764 microRNAs known for Rattus Norvegicus analyzed? These may not be expressed in the ischemic stroke model in the brain regions analyzed and the obtained network analysis results may not be present in the brain regions. Please include the rationale for including all microRNAs or include some more information in the discussion regarding this.

Authors:

In accordance with the Reviewer’s recommendation, changes were added in the text of the Manuscript as a limitation of our study (lines 417-418 in Mark-up Copy_R1).

Reviewer 2:

  1. Why does targetscan yield significantly higher number of interactions compared to the other techniques used? Please include any limitations with this method.

Authors:

We are grateful to the Reviewer for the comment. In accordance with the Reviewer's recommendation, we stated determining the interaction using only computational methods as a limitation of our study (lines 415-417 in Mark-up Copy_R1).

The results of various programs for predicting microRNA-mRNA interactions, and especially microRNA-circRNA interactions, generally show quite poor overlap, which is a discussed issue (see, for example, PMID: 30919506, PMID: 20805242). Most bioinformatics approaches used today produce a large number of both false positives and false negatives. TargetScan predicts microRNA targets by searching for pre-annotated 8-, 7-, and 6-nucleotide sites that correspond to the seed region of each microRNA [https://www.targetscan.org/vert_80/]. At this stage, false positive results can occur. Under these conditions, it is preferable to simultaneously use several programs implementing different approaches to predicting RNA interactions. We use precisely such approach in our study.

Reviewer 2:

  1. Line 292- Most of them (403 DECs) showed decreased expression. Please discuss a hypothesis, referring to previous studies, why this might be the case after an ischemic stroke and how this is important to the transcriptomic changes observed after stroke. This provides more biological relevance to the results observed in this study.

Authors:

We are grateful to the Reviewer for the comment. In accordance with the Reviewer’s recommendation, changes were added in the text of the Manuscript (lines 308-313 in Mark-up Copy_R1).
